# Social disconnectedness, economic outcomes, and the role of pre-existing mental health conditions: A population-based cohort study

Lisbeth Mølgaard Laustsen[1,2*], Mathias Lasgaard[2,3], Niels Skipper[4], Danni Chen[1], Jaimie L. Gradus[5,6], Marie Stjerne Grønkjær[7], Oleguer Plana-Ripoll[1,8*]

**1** Department of Clinical Epidemiology, Aarhus University and Aarhus University Hospital, Aarhus, Denmark, **2** DEFACTUM – Public Health Research, Central Denmark Region, Aarhus, Denmark, **3** Department of Psychology, University of Southern Denmark, Sønderborg, Denmark, **4** Department of Economics and Business Economics, Business and Social Sciences, Aarhus University, Aarhus, Denmark, **5** Boston University School of Public Health, Boston, Massachusetts, Unites States of America, **6** Boston University School of Medicine, Boston, Massachusetts, Unites States of America, **7** Center for Clinical Research and Prevention, Copenhagen University Hospital – Bispebjerg and Frederiksberg, Capital Region of Denmark, Denmark, **8** National Centre for Register-based Research, Department of Public Health, Aarhus University, Aarhus, Denmark

\* lml@clin.au.dk (LML); opr@clin.au.dk (OP-R)

## Abstract

Despite the growing recognition of social disconnectedness as a critical public health issue, significant gaps in the existing evidence remain regarding its associated economic outcomes. This study aimed to examine differences in economic outcomes according to social disconnectedness and explore variation based on sex, age, and pre-existing mental health conditions. We conducted a cohort study linking data on social disconnectedness (loneliness, social isolation, low social support, and a composite measure) from the Danish National Health Survey in 2013 and 2017 (n = 158,970) with register data on economic outcomes (healthcare costs, wage income, and transfer payments) in the following calendar year. We used linear regression to estimate mean differences including sex- and age-stratified analyses and an analysis of interaction with pre-existing mental health conditions. Individuals who were socially disconnected had on average annual excess healthcare costs of €561 to €1,674, lower wage income of €7,207 to €16,295, and excess transfer payments of €2,845 to €8,977. When extrapolated, excess healthcare costs and transfer payments associated with the composite measure corresponded to 7.1% of all transfer payments and 9.0% of all healthcare costs. These economic differences varied according to sex and age and were greater among individuals with pre-existing mental health conditions. These findings can provide insights into economic disparities and may inform initiatives to optimize healthcare resource allocation and enhance workforce participation.

**Data availability statement:** Data presented in this study were obtained from Danish registries and regions participating in the Danish National Health Survey. Owing to data protection rules, individual-level data may not be shared publicly. Other researchers who fulfil the requirements set by the data providers may gain access to the data through Statistics Denmark, the Danish Health Data Authority, and/or the Danish regions (Central Denmark Region, North Denmark Region, Region Zealand, and Capital Region of Denmark). A preregistered analysis plan and all statistical code from the main analysis is available at Open Science Framework (https://osf.io/mb2ua/).

**Funding:** The project is supported by the Graduate School of Health at Aarhus University in the form of a PhD fellowship [to LML] and the Lundbeck Foundation in the form of a fellowship [R345-2020-1588 to OP-R]. This project was also supported by the Independent Research Fund Denmark in the form of grants [1030-00085B and 2066-00009B to OP-R]. The funders had no role in study design, data collection and analysis, decision to publish, or preparation of the manuscript.

**Competing interests:** The authors have declared that no competing interests exist.

## Introduction

Social disconnectedness is increasingly recognized as a critical public health issue with implications extending beyond mental health and wellbeing [1]. There is strong evidence that individuals who are socially disconnected are at increased risk of mental health conditions [2], dementia [3], and coronary heart disease [4], and recent studies have shown increased incidence rates for a broad range of medical conditions [5,6]. These medical conditions may necessitate frequent medical visits, hospitalizations, and the need for specialized medications and treatments, thereby possibly elevating healthcare costs for individuals who are socially disconnected [7]. Conversely, the absence of support from strong social networks may be a hindrance to the utilization of needed healthcare services, potentially leading to reduced healthcare costs, but increased income loss due to sickness absence and increased transfer payments such as disability pension [8,9]. At the same time, functional limitations among individuals with medical conditions may also subsequently lead to withdrawal from or reduced quality of existing social networks [10].

Significant gaps and limitations remain in the evidence concerning economic outcomes associated with social disconnectedness [11]. First, prior studies have often focused on single aspects of social disconnectedness, specific patient groups, and/or specific economic outcomes (for an overview of previous studies, see S1 Table). Second, existing studies have predominantly included older populations, whereas adolescents and young adults have received less attention. Third, although few studies have examined sex and age differences, such differences are likely due to demographic differences in disease patterns, healthcare utilization, and income. Given the strong association between mental health conditions and social disconnectedness [12] along with the documented excess costs among people with mental health conditions [13] it is also imperative to explore how mental health conditions might influence economic outcomes associated with social disconnectedness. Accordingly, a more comprehensive overview of the economic differences linked to social disconnectedness is needed. Quantifying these differences could be of relevance for economic considerations of preventative efforts and provide insight into the potential for developing tailored interventions.

The aim of this study was thus to provide a comprehensive overview of differences in healthcare costs, wage income, and transfer payments (e.g., unemployment benefits, subsidized employment, and disability pension) according to three distinct aspects of social disconnectedness (loneliness, social isolation, and low social support), as well as a composite measure of these factors. Additionally, the study aimed to explore these economic differences according to sex and age and assess interaction with pre-existing mental health conditions.

## Materials and methods

### Study design and population

This study was designed as a cohort study of participants from the Danish National Health Survey, which was conducted in five regional stratified random samples and

one national random sample [14]. Based on the availability of the applied survey items, we included 33,285 survey participants from one region collected during January 30 to April 30, 2013 (Central Denmark Region) and 129,319 survey participants from four regions collected during February 3 to April 30, 2017 (Central Denmark Region, North Denmark Region, Region Zealand, and Capital Region of Denmark). In total, the initial study population encompassed 162,604 individuals with a response rate of 57.5%. Subsequently, we applied unique identification numbers from the Danish Civil Registration System [15] to link with individual-level information in the national registers on public healthcare utilization, wage income, transfer payments, and covariates. We included 3,426 responses (2.1%) which were provided by individuals who by chance participated in both 2013 and 2017 but excluded 107 individuals (0.07%) with no register linkage at the time of survey participation, leaving a potential sample of 162,497 individuals. We subsequently excluded 1,071 (0.7%) individuals who emigrated and 2,456 (1.5%) individuals who died before the end of the calendar year following survey participation (2014 or 2018) for which their annual measures were incomplete. The analytical sample thus comprised 158,970 individuals. A flow-chart illustrating the definition of the study population is provided in S1 Fig. To address non-response and selection probabilities, we applied inverse probability weights that have been made available with the survey data and are constructed by Statistics Denmark with a model-based calibration approach based on information from the national registers [14].

## Ethics

The study was registered with the Danish Data Protection Agency at Aarhus University (No 2016-051-000001-2587) and approved by Statistics Denmark and the Danish Health Data Authority. According to Danish law, informed consent or ethical approval is not required for register-based studies in Denmark. For survey participants, information about the survey was provided to potential participants in writing. All survey participants were informed that participation was voluntary and that their survey data would be linked to the registers for research purposes. The respondents' full or partial completion of the survey constituted implied consent. Linking the survey data and register data was done by Statistics Denmark. All data were pseudonymized and not recognizable at an individual level and analysed on the secure platform of Statistics Denmark.

## Social connections

Loneliness was assessed in the Danish National Health Survey with the Danish version of the Three-Item Loneliness Scale [16,17], providing a score ranging from 3 to 9 with higher scores indicating greater loneliness. In 2017, the third item was slightly rephrased to enhance correspondence with the definition of loneliness, but the scale has demonstrated good internal consistency at both time points [18]. Corresponding to the most conservative dichotomisation of severe loneliness applied in the literature [19], we classified a score of 7 or higher as indicating loneliness. Social contacts were assessed in the survey by quantifying different areas of social contact with inspiration from the Berkman-Syme Social Network Index [20]. Four indicators of limited social contact were used to measure social isolation providing a score ranging from 0 to 4: whether an individual i) was living alone, ii) was unemployed and not enrolled in education, iii) had less than monthly contact with friends, and iv) had less than monthly contact with family outside of the household. We classified a score of 3 or higher as indicating social isolation. Social support was assessed in the survey as perceived emotional support with inspiration from the MOS Social Support Instrument [21]. The single-item *"Do you have someone to talk to if you have problems or need for support?"* with four response options: *"Yes, always"*; *"Yes, mostly"*; *"Yes, sometimes"*; and *"No, never or almost never"* was applied. We classified answers in the two last-mentioned response options as indicating low social support. To capture both structural and functional aspects of social disconnectedness, we additionally constructed a composite measure of either loneliness, social isolation, or low social support.

## Economic outcomes

Healthcare costs included public reimbursements of healthcare services delivered by general practitioners and medical specialists from the Danish National Health Service Register [22], public cost averages for hospital-based services from

the DRG National Patient Register and the DRG Psychiatric Patient Register, and public subsidies for redeemed prescriptions from the Danish National Prescription Registry [23]. For each individual, total healthcare costs were calculated by summing the costs of primary healthcare services, hospital-based services, and subsidies for redeemed prescriptions. Wage income and transfer payments were assessed using information from the Income Statistics Register [24] on the annual primary taxable wage income (i.e., excluding transfer payments) and a range of transfer payments (e.g., unemployment benefits, subsidized employment, and disability pension) which are primarily publicly funded. All economic outcomes were assessed in the calendar year following survey participation (2014 or 2018), and outcomes from 2014 were weighted to represent values in 2018 of Danish kroner using the gross domestic product deflator from the World Bank. Subsequently, we converted from Danish kroner to euros using the average exchange rate from the European Central Bank in 2018 (1 euro equalling 7.47 Danish kroner). Further data documentation is available at Open Science Framework (https://osf.io/f2vm5).

## Covariates

Sex (registered legal sex), age when participating in the Danish National Health Survey, and country of birth (Denmark and Greenland vs. abroad) were obtained from the Danish Civil Registration System [15]. Information on hospital-diagnosed mental health conditions in 18 years preceding survey participation was obtained from the Danish Psychiatric Central Research Register [25], which since Jan 1, 1995, contains data on both outpatient services in psychiatric departments and admissions to inpatient and emergency psychiatric departments with the International Classification of Diseases, 10th revision (ICD-10) [26]. Organic disorders (ICD-10: F00–F09) and intellectual disabilities (ICD-10: F70–F79) were not included in the applied definition due to considerations regarding distinctive characteristics of these patient groups, but all other psychiatric diagnoses (ICD-10: F10–F69 & F80–F99) were included.

## Statistical analysis

To avoid introducing selection bias [27], multiple imputation by chained equations were conducted to include 19,675 (12.4%) individuals with partially missing survey data (S1 Text). Means, standard deviations (SDs), and proportions were computed to describe the baseline characteristics of the cohort. All estimates were calculated with inverse probability of participation weights and pooled multiple imputed data using Rubin's Rules.

All analyses were performed separately for each indicator of social disconnectedness (loneliness, social isolation, low social support, and the composite measure). We applied linear regression models with Taylor-linearized variance estimation and 95% confidence intervals (CIs) to compare each economic outcome between individuals who were socially disconnected at baseline and individuals who were not. Adjustments were made for sex, age, year of survey participation, and country of birth. The total of each economic outcome was estimated applying marginal standardization to average over the distribution of the covariates for individuals who were socially disconnected. Additionally, we extrapolated the results to the entire population in the corresponding Danish regions with the inverse probability weights and calculated the attributable share of total healthcare costs and transfer payments, respectively. As the first sensitivity analysis, we applied quantile regression to explore whether economic differences according to social disconnectedness were consistently found when modelling the first, second, and third quartile distribution of each economic outcome instead of the mean [28]. We also repeated the main analysis stratified based on sex and age groups in 10-year intervals. Furthermore, we analysed interaction with pre-existing hospital-diagnosed mental health conditions by estimating the deviation from additivity, also known as the interaction contrast. As the second sensitivity analysis, we repeated the interaction analysis with a broader definition of pre-existing mental health conditions additionally including self-reported information, redeemed prescriptions of psychopharmaceuticals, and consultations with private practicing psychiatrists. Additionally, we repeated the interaction analysis stratified by sex. Details are provided in S1 Text.

Statistical analyses were conducted in Stata version 18.0 using the svy and mi suite of commands on register data provided by Statistics Denmark between June 3 and September 8, 2022. A preregistered analysis plan and the code used for data management and statistical analysis are available at Open Science Framework (https://osf.io/mb2ua).

## Results

The analytical sample consisted of 158,970 survey participants with a mean age of 48.1 years (SD 18.9) at baseline, of which 86,002 (50.7%) were women, 11,678 (11.8%) were born abroad, and 32,544 (21.6%) participated in the survey in 2013. The number of individuals classified as lonely, socially isolated, and with low social support was 9,460 (7.5%), 4,321 (3.3%), and 20,807 (14.8%), respectively. Individuals in one of these three groups were more likely to have a pre-existing hospital-diagnosed mental health condition compared to individuals in none of these groups (13.9 to 26.0% vs. 5.7%). Additional baseline characteristics according to social disconnectedness are shown in Table 1.

The total and the differences in annual healthcare costs, wage income, and transfer payments according to each indicator of social disconnectedness are provided in Fig 1. Individuals who were lonely compared to individuals who were not had on average excess annual healthcare costs of €1,674 (95% CI, €1,343 to €2,005) in 2018 prices, whereas this was €1,515 (95% CI, €938 to €2,093) for individuals who were socially isolated, €561 (95% CI, €397 to €725) for individuals with low social support, and €828 (95% CI, €671 to €985) for individuals with any combination of these factors. The greatest part of these excess healthcare costs was attributable to psychiatric inpatient services, followed by somatic inpatient services and subsidised prescriptions. When extrapolating these results to the entire population within the included Danish regions, excess healthcare costs associated with loneliness, social isolation, low social support, and any combination of these factors corresponded to respectively 5.5%, 2.2%, 3.6%, and 7.1% of all healthcare costs.

With regard to income, individuals who were lonely compared to individuals who were not had on average €13,466 (95% CI, €12,809 to €14,123) lower annual wage income in 2018 prices, whereas this was €16,295 (95% CI, €15,467 to €17,124) for individuals who were socially isolated, €7,207 (95% CI, €6,711 to €7,702) for individuals with low social support, and €9,253 (95% CI, €8,818 to €9,688) for individuals with any combination of these factors. This wage income gap was partly counterbalanced by excess annual transfer payments of on average €5,453 (95% CI, €5,171 to €5,736) in 2018 prices for individuals who were lonely, €8,977 (95% CI, €8,583 to €9,372) for individuals who were socially isolated, €2,845 (95% CI, €2,666 to €3,025) for individuals with low social support, and €3,875 (95% CI, €3,713 to €4,037) for individuals with any combination of these factors. This made the total annual income gap for individuals who were lonely, socially isolated, with low social support, and any combination of these factors respectively €8,013, €7,318, €4,361, and €5,379. When extrapolating these results to the entire population within the included Danish regions, excess transfer payments associated with loneliness, social isolation, low social support, and any combination of these factors corresponded to respectively 4.8%, 3.5%, 5.0%, and 9.0% of all transfer payments.

In the sensitivity analysis of the first, second, and third quartile of each outcome, economic differences in the same direction were consistently found but the estimated difference varied depending on the quartile, indicator, and economic outcome (S2 Fig).

In the sex-stratified analysis shown in Fig 2, women compared to men had higher excess healthcare costs for loneliness and low social support, but lower for social isolation. These differences were especially driven by higher excess costs of somatic outpatient services among women for loneliness and low social support, and higher excess costs of inpatient psychiatric services among men for social isolation. With regard to income, men compared to women had a greater wage income gap for all indicators of social disconnection, especially social isolation (€20,442 [95% CI, €19,186 to €21,698] vs. €11,059 [95% CI, €10,104 to €12,014]). This sex difference in the income gap was not counterbalanced by higher excess transfer payments among men compared to women for low social support, whereas this was partly the case for loneliness and especially social isolation (€9,877 [95% CI, €9,323 to €10,432] vs. €7,785 [95% CI, €7,241 to €8,329]).

**Table 1. Baseline characteristics of the cohort in four regions of Denmark, 2013 and 2017.** Missing data was imputed using multiple impu-tation by chained equations. Absolute numbers are unweighted, whereas means, standard deviations, and percentages are weighted based on register data to represent the population of the included regions in 2013 and 2017. Note that loneliness, social isolation, and low social support are not mutually exclusive; therefore, the percentages in the top row do not sum up to 1.

| | Lonely: N=9,460 (7.5%) | Socially isolated: N=4,321 (3.3%) | Low social support: N=20,807 (14.8%) | Neither lonely, socially isolated, or low social support: N=131,671 (80.3%) |
|---|---|---|---|---|
| Age, mean (SD) | 42.7 (24.3) | 63.0 (27.2) | 47.9 (24.5) | 48.1 (22.8) |
| Women, N (%) | 5,644 (55.3) | 1,997 (44.2) | 10,204 (45.2) | 72,004 (51.4) |
| Survey participation in 2013 as opposed to 2017, N (%) | 1,199 (13.6) | 820 (19.2) | 4,127 (20.2) | 27,451 (22.3) |
| Born abroad, N (%) | 1,483 (22.4) | 443 (15.1) | 2,843 (20.8) | 8,068 (9.8) |
| The Three-Item Loneliness Scale, mean (SD) | 7.8 (1.2) | 5.5 (2.9) | 5.4 (2.6) | 3.6 (1.2) |
| The adapted Social Isolation Index | | | | |
| Living alone, N (%) | 3,527 (40.1) | 3,226 (78.6) | 5,978 (33.1) | 20,589 (18.8) |
| Out of employment and not enrolled in education N (%) | 4,304 (43.9) | 4,086 (93.2) | 8,485 (39.4) | 44,871 (30.4) |
| Less than monthly contact with friends, N (%) | 2,376 (24.6) | 3,155 (72.6) | 4,146 (20.3) | 7,028 (5.1) |
| Less than monthly contact with family N (%) | 1,829 (20.1) | 2,981 (68.3) | 3,558 (18.6) | 5,997 (5.0) |
| The social support item | | | | |
| Social support never or almost never available, N (%) | 1,881 (20.4) | 909 (23.1) | 6,447 (32.2) | 0 (0) |
| Social support sometimes available, N (%) | 3,168 (33.0) | 981 (23.3) | 14,361 (67.8) | 0 (0) |
| Social support mostly available, N (%) | 2,680 (28.1) | 1,125 (25.1) | 0 (0) | 39,101 (30.2) |
| Social support always available, N (%) | 1,731 (18.5) | 1,306 (28.5) | 0 (0) | 92,570 (69.8) |
| Hospital-diagnosed mental health condition, N (%) | 2,263 (26.0) | 761 (20.9) | 2,483 (13.9) | 6,349 (5.7) |
| Self-reported mental health condition, N (%) | 4,248 (46.0) | 1,245 (32.9) | 5,380 (27.7) | 16,236 (13.2) |
| Psychopharmacological redemption, N (%) | 4,832 (48.8) | 2,184 (51.6) | 7,797 (36.6) | 31,778 (22.9) |
| Consultation with a private practicing psychiatrist, N (%) | 1,621 (17.4) | 558 (14.2) | 2,062 (10.5) | 5,586 (4.5) |

Consequently, men compared to women had a greater total income gap for all indicators of social disconnection (€5,126 to €10,838 vs. €3,274 to €6,027).

In the age-stratified analyses shown in Fig 3, we found excess healthcare costs for individuals who were socially disconnected for all indicators and for all age groups, except for individuals in the oldest age group (≥76 years) who were socially isolated. In general, the contribution to excess healthcare costs from somatic services increased with age, whereas the contribution from psychiatric services decreased with age. However, individuals in the oldest age group (≥76 years) had reduced costs of somatic outpatient services according to all indicators of social disconnectedness. The highest excess healthcare costs were found for individuals aged 46–55 years. With regard to wage income and transfer payments, we found lower wage income and excess transfer payments for all age groups with the greatest wage income gap for individuals aged 36–55 years and the highest excess transfer payments for individuals aged 46–55 years.

In the interaction analysis shown in Fig 4, we found that the differences in healthcare costs were above that expected based on additive interaction for each indicator of social disconnectedness and a pre-existing hospital-diagnosed mental health condition with deviations ranging from €251 to €1,732. With regard to wage income and transfer payments, we found small deviations above that expected based on additivity, except for wage income according to loneliness, where we found less than expected based on additivity. However, these deviations from additivity were estimated with consider-able uncertainty, and the differences in healthcare costs according to social isolation and low social support and in wage income according to loneliness could also be equal to or lower than expected based on additive interaction. Similar trends regarding deviations from additive interaction were found when using a broader definition of mental health conditions

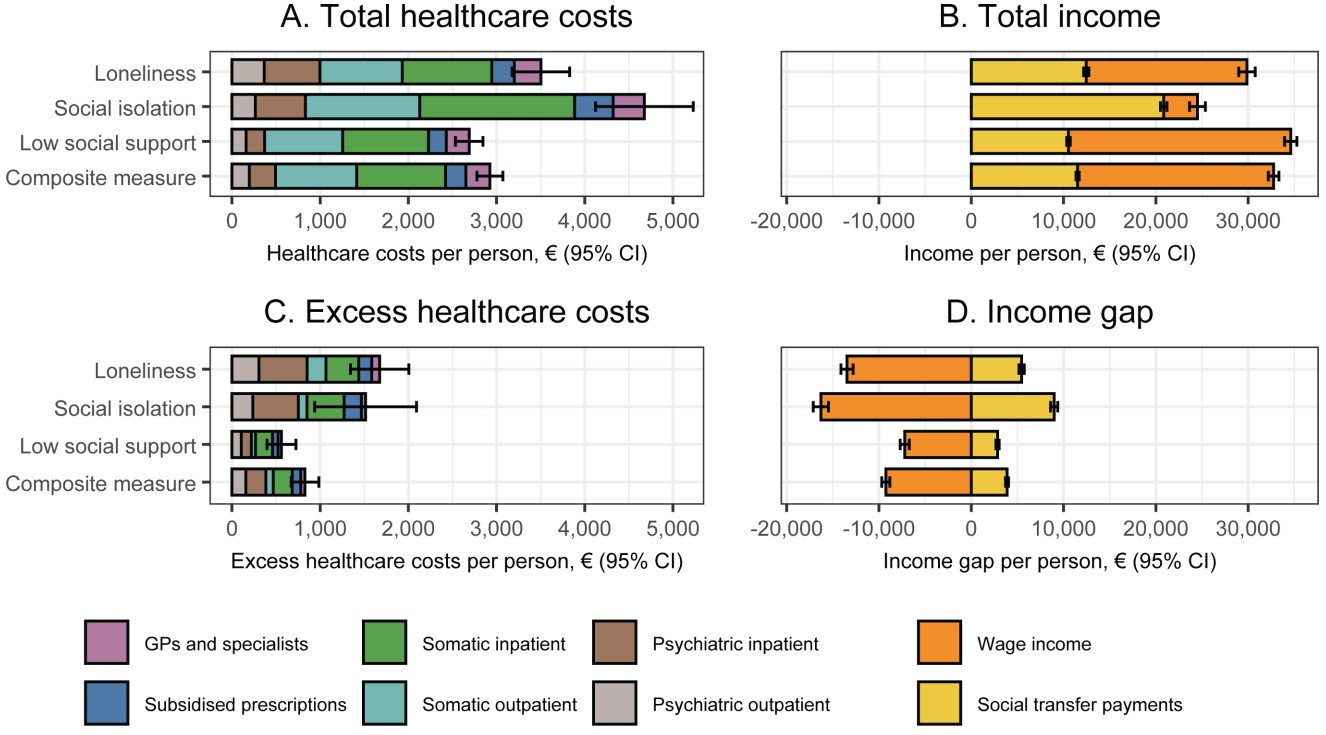

**Fig 1. Total and differences in annual healthcare costs, wage income, and transfer payments according to each indicator of social disconnectedness in four regions of Denmark, 2014 & 2018.** CI: Confidence interval; GPs: General practitioners. The colours indicate the contributions of different cost categories. Missing data was imputed using multiple imputation by chained equations, and the results are weighted based on register data to represent the population of the included regions in 2013 and 2017. The total estimates are calculated using marginal standardization to the distribution of covariates among individuals with each indicator of social disconnectedness (loneliness, social isolation, low social support, and the composite measure, respectively). The estimates represent values in 2018 and are adjusted for sex, age (included as a natural cubic spline with five knots), year of survey participation, and country of birth. Estimates are provided in S2 Table.

(S3 Fig). The sex-stratified interaction analysis showed predominantly greater deviations from additivity among men compared to women, and healthcare costs were conversely below that expected based on additive interaction among women with low social support and a pre-existing hospital-diagnosed mental health condition (S4 Fig).

## Discussion

To the best of our knowledge, this study provides the most comprehensive examination to date of differences in economic outcomes according to social disconnectedness and the interaction with pre-existing mental health conditions. Based on 158,970 participants from the Danish National Health Survey linked with register data, we found that individuals with three different aspects of social disconnectedness, as well as a composite measure of these factors, had on average €561 to €1,674 in excess annual healthcare costs, €7,207 to €16,295 in lower annual wage income, and €2,845 to €8,977 in excess annual transfer payments. When extrapolated to the included Danish regions, the results suggest that excess healthcare costs and transfer payments associated with the composite measure corresponded to 7.1% of all transfer payments and 9.0% of all healthcare costs. However, our results should be interpreted with caution as they do not represent causal estimates. Furthermore, we found that these economic differences varied according to sex and age and were generally greater than expected among individuals with both social disconnectedness and a pre-existing hospital-diagnosed mental health condition.

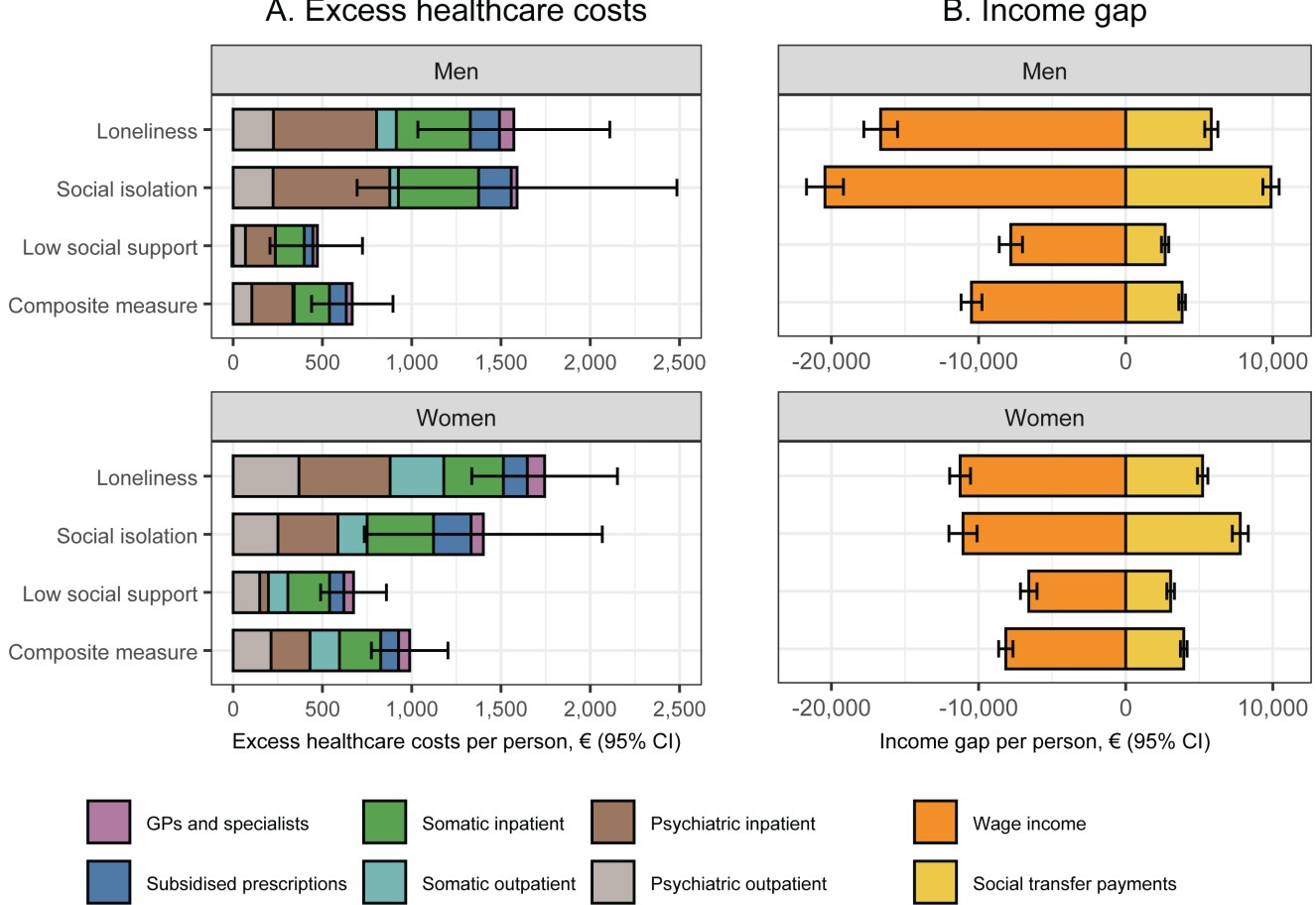

**Fig 2. Sex-specific differences in annual healthcare costs, wage income, and transfer payments according to each indicator of social discon-nectedness in four regions of Denmark, 2014 & 2018.** CI: Confidence interval; GPs: General practitioners. The colours indicate the contributions of different cost categories. Missing data was imputed using multiple imputation by chained equations, and the results are weighted based on register data to represent the population of the included regions in 2013 and 2017. The estimates represent values in 2018 and are adjusted for age (included as a natural cubic spline with five knots), year of survey participation, and country of birth. Estimates are provided in S3 Table.

## Comparison to prior findings

Our findings of excess healthcare costs and transfer payments for individuals who were socially disconnected are in line with several previous studies [7–9]. Nevertheless, a Dutch population-based study involving 341,376 individuals found, contrary to our results, reduced costs of somatic inpatient and outpatient services among individuals who were lonely [7]. A major difference in this comparison is that our findings are based on models with basic demographic adjustments to obtain descriptive estimates, whereas this study included adjustments for several variables that we conceptualized as potential mediators (i.e., socioeconomic and lifestyle-related factors, self-perceived health, and psychological distress). In a Swedish population-based study based on 53,920 individuals aged 20–64 years, they reported a higher risk of receiving disability pension for individuals who were socially isolated, except for men aged 40–64 years [8]. Similarly, a Dutch study predicted that fewer workers with high social support would take long-term sickness leave [9]. To our knowledge, our study is the first to explore the interaction with mental health conditions on economic outcomes.

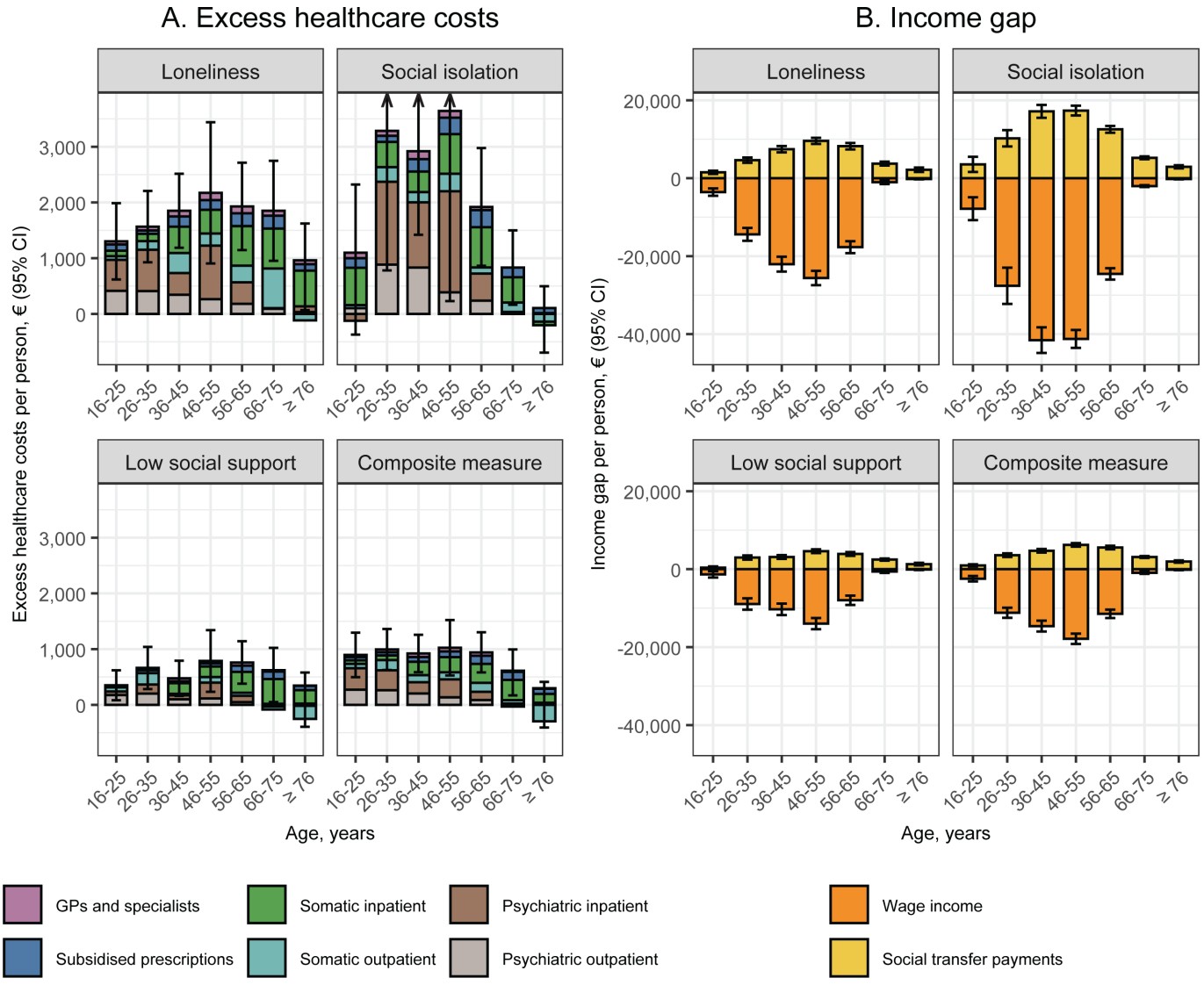

**Fig 3. Age-specific differences in annual healthcare costs, wage income, and transfer payments according to each indicator of social disconnectedness in four regions of Denmark, 2014 & 2018.** CI: Confidence interval; GPs: General practitioners. The colours indicate the contributions of different cost categories. Missing data was imputed using multiple imputation by chained equations, and the results are weighted based on register data to represent the population of the included regions in 2013 and 2017. The estimates represent values in 2018 and are adjusted for sex, age (included as a natural cubic spline with three knots), year of survey participation, and country of birth. Estimates are provided in S4 Table.

### Strength and limitations

This study benefitted from utilizing rich survey data, thus allowing us to assess three distinct aspects of social disconnectedness with a validated measure of loneliness [17] and a measure of social isolation created with inspiration from a well-known index [20]. Another strength is the assessment of several economic outcomes—a range of healthcare costs, wage income, and transfer payments—using high-quality administrative register data with nearly complete records. The study also benefitted from studying a population-based sample allowing us to explore variation across subgroups, while applying inverse probability weights that have been shown to account well for primary healthcare utilization [29] and mental health conditions [30]. Not least, multiple imputation was applied to avoid introducing selection bias when dealing with missing survey data.

**Fig 4. Interaction between each indicator of social disconnectedness and pre-existing hospital-diagnosed mental health conditions on annual healthcare costs, wage income, and transfer payments in four regions of Denmark, 2014 & 2018.** CI: Confidence interval; GPs: General practitioners. The colours indicate the contributions of different cost categories. Missing data was imputed using multiple imputation by chained equations, and the results are weighted based on register data to represent the population of the included regions in 2013 and 2017. The estimates represent values in 2018 and are adjusted for sex, age (included as a natural cubic spline with five knots), year of survey participation, and country of birth. Estimates are provided in S5 Table.

However, the study also has important limitations. First, the study design does not allow inferences regarding whether social disconnectedness is a cause or an effect of the observed economic differences, or whether these differences are in fact due to other confounding factors. Second, the measure of social support and social isolation have not been validated, and the measure of social isolation did not include associational activities and voluntary work, whereas the measure of social support focused solely on perceived emotional support [31]. Third, despite the application of inverse probability weights based on administrative registers, the representativeness of the study participants cannot be confirmed with regard to factors not recorded in the registers. Notably, our finding that lower wage income was partially counterbalanced by excess transfer payments illustrates the relatively high level of welfare benefits available in Denmark. Accordingly, these findings may have limited generalizability for countries with different healthcare systems, economic structures, and social policies.

## Potential explanations

Our findings of excess healthcare costs point towards three different reasonings. First, the higher healthcare costs may be attributed to a higher prevalence of medical conditions among individuals who are socially disconnected, consistent with numerous studies [2–5]. Second, individuals who are socially disconnected may face a greater severity of medical conditions, as observed for depression [32] and multimorbidity [33]. Third, the results could be explained by higher cost of treatments for comparable medical conditions among individuals who are socially disconnected (e.g., due to complications or readmissions). For instance, an association between low social support and poor self-management behaviours has been identified [34]. Our findings of lower wage income and excess transfer payments may be attributed to both the proportion of individuals who are employed and receiving transfer payments, as well as the average wage income among those employed and the average transfer payments among recipients. The former indicates differences due to health disparities, as discussed above, while the latter could reflect broader social inequalities, potentially caused by a wide range of factors.

## Conclusions

Taking all our findings into account, one explanation for our results is a positive influence of social connections on health, e.g., through stress-buffering and promotion of healthy behaviours [35]. Concurrently, another explanation for our findings is that functional limitations among individuals with medical conditions have subsequently led to changes in social contact. To distinguish cause and effect, future studies with high-quality measures of social disconnectedness over time are needed. Nevertheless, these findings of substantial economic differences between socially connected and disconnected population groups may help elucidate the potential economic benefits of preventative efforts. Our findings may thus serve as a crucial first step towards developing interventions aimed at optimizing healthcare resources and enhancing workforce participation. Not least, our findings of variation in these economic outcomes for different subgroups could inform tailored interventions and help the generation of hypotheses regarding the underlying causes.

## Supporting information

**S1 Text. Supplementary methods.**
(PDF)

**S1 Fig. Flowchart depicting the study population in four regions of Denmark, 2013 and 2017.**
(PDF)

**S2 Fig. The first, second, and third quartile of annual healthcare costs, wage income, and transfer payments according to each indicator of social disconnectedness in four regions of Denmark, 2014 & 2018.**
(PDF)

**S3 Fig. Sensitivity analysis applying a broader definition of pre-existing mental health conditions in the analysis of interaction with each indicator of social disconnectedness on annual healthcare costs, wage income, and transfer payments in four regions of Denmark, 2014 & 2018.**
(PDF)

**S4 Fig. Sex-specific interaction between each indicator of social disconnectedness and pre-existing hospital-diagnosed mental health conditions on annual healthcare costs, wage income, and transfer payments in four regions of Denmark, 2014 & 2018.**
(PDF)

**S1 Table. Overview of prior studies assessing social disconnectedness and economic outcomes.**
(PDF)

**S2 Table. Total and differences in healthcare costs, wage income, and transfer payments according to each indicator of social disconnectedness in four regions of Denmark, 2014 & 2018.**
(PDF)

**S3 Table. Sex-specific differences in annual healthcare costs, wage income, and transfer payments according to each indicator of social disconnectedness in four regions of Denmark, 2014 & 2018.**
(PDF)

**S4 Table. Age-specific differences in annual healthcare costs, wage income, and transfer payments according to each indicator of social disconnectedness in four regions of Denmark, 2014 & 2018.**
(PDF)

**S5 Table. Interaction between each indicator of social disconnectedness and pre-existing hospital-diagnosed mental health conditions on annual healthcare costs, wage income, and transfer payments in four regions of Denmark, 2014 & 2018.**
(PDF)

## Acknowledgments

The Central Denmark Region Health Survey was conducted and funded by the Central Denmark Region. The North Denmark Region Health Survey was conducted and funded by the North Denmark Region. The Danish Capital Region Health Survey was conducted and funded by the Capital Region. The Region Zealand Health Survey was conducted and funded by Region Zealand.

## Author contributions

**Conceptualization:** Lisbeth Mølgaard Laustsen, Mathias Lasgaard, Niels Skipper, Oleguer Plana-Ripoll.

**Data curation:** Lisbeth Mølgaard Laustsen, Danni Chen, Oleguer Plana-Ripoll.

**Formal analysis:** Lisbeth Mølgaard Laustsen.

**Funding acquisition:** Lisbeth Mølgaard Laustsen, Oleguer Plana-Ripoll.

**Investigation:** Mathias Lasgaard, Marie Stjerne Grønkjær.

**Methodology:** Lisbeth Mølgaard Laustsen, Mathias Lasgaard, Niels Skipper, Jaimie L. Gradus, Marie Stjerne Grønkjær, Oleguer Plana-Ripoll.

**Supervision:** Oleguer Plana-Ripoll.

**Validation:** Danni Chen.

**Visualization:** Lisbeth Mølgaard Laustsen.

**Writing – original draft:** Lisbeth Mølgaard Laustsen.

**Writing – review & editing:** Mathias Lasgaard, Niels Skipper, Danni Chen, Jaimie L. Gradus, Marie Stjerne Grønkjær, Oleguer Plana-Ripoll.

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
