## [Decision Letter · Decision Letter 0]

13 Feb 2025

PMEN-D-24-00562

Social disconnectedness, economic outcomes, and the role of pre-existing mental disorders: A population-based cohort study

PLOS Mental Health

Dear Dr. Laustsen,

Thank you for submitting your manuscript to PLOS Mental Health. After careful consideration, we feel that it has merit but does not fully meet PLOS Mental Health’s publication criteria as it currently stands. Therefore, we invite you to submit a revised version of the manuscript that addresses the points raised during the review process.

This study is a rigorous examination of the associations between social disconnectedness and economic outcomes using a sufficiently large sample and controlling for gender, age, and pre-existing mental disorders. Because this is a cross-sectional study, it is impossible to determine causal relationships, but this study is worthy of publication in this journal. Please respond appropriately to the reviewer's comment (clearly state the internal consistencies for loneliness and social isolation) and the editor's suggestion (describe the results clearly and understandably).

We look forward to receiving your revised manuscript.

Kind regards,

Hirokazu Taniguchi, Ph.D.

Academic Editor

PLOS Mental Health

Journal Requirements:

https://journals.plos.org/mentalhealth/s/figures 

https://journals.plos.org/mentalhealth/s/figures#loc-file-requirements 

2. Please provide an Author Summary. This should appear in your manuscript between the Abstract (if applicable) and the Introduction, and should be 150–200 words long. The aim should be to make your findings accessible to a wide audience that includes both scientists and non-scientists. Sample summaries can be found on our website under Submission Guidelines:

https://journals.plos.org/globalpublichealth/s/submission-guidelines#loc-parts-of-a-submission.

Additional Editor Comments (if provided):

There do not appear to be any significant problems that need to be corrected; there is one point where the author(s) could review the description for appropriateness. The description on lines 301-302 of the Results section seems vague and difficult for the reader to understand. According to the results in Figure 4 and Table S5, the confidence intervals of the deviation from additivity for social isolation and low social support include zero and negative values. Thus, the healthcare costs for the two indicators could be equal to or lower than expected based on the additive interaction. It would be better to state these things clearly.

In addition, the supplementary tables in the footnotes to Figures 1 through 4 are incorrectly numbered. For example, the estimates in Figure 1 are reported in Table S2, not Table S3. Please correct these errors.

Reviewers' comments:

Reviewer's Responses to Questions

**Comments to the Author**

1. Does this manuscript meet PLOS Mental Health’s publication criteria ? Is the manuscript technically sound, and do the data support the conclusions? The manuscript must describe methodologically and ethically rigorous research with conclusions that are appropriately drawn based on the data presented.

Reviewer #1: Yes

2. Has the statistical analysis been performed appropriately and rigorously?

Reviewer #1: Yes

3. Have the authors made all data underlying the findings in their manuscript fully available (please refer to the Data Availability Statement at the start of the manuscript PDF file)?

Reviewer #1: Yes

4. Is the manuscript presented in an intelligible fashion and written in standard English?

Reviewer #1: Yes

5. Review Comments to the Author

Reviewer #1: Thank you for the opportunity to review this manuscript. The authors' study examines the differences in economic outcomes associated with social disconnectedness and its interaction with pre-existing mental disorders. The manuscript is objectively and clearly written, with a well-organized structure. The analysis is detailed and rigorous. Notably, the authors provide a comprehensive analysis of the missing data mechanism, an aspect rarely addressed in published articles. I have only one question for clarification: How is the internal consistency of the measures for social contacts and social support? Which test did you use to evaluate it?

6. PLOS authors have the option to publish the peer review history of their article (what does this mean? ). If published, this will include your full peer review and any attached files.

**Do you want your identity to be public for this peer review?** For information about this choice, including consent withdrawal, please see our Privacy Policy .

Reviewer #1: No

---

## [Decision Letter · Decision Letter 1]

26 Mar 2025

Social disconnectedness, economic outcomes, and the role of pre-existing mental disorders: A population-based cohort study

PMEN-D-24-00562R1

Dear Ms Laustsen,

We are pleased to inform you that your manuscript 'Social disconnectedness, economic outcomes, and the role of pre-existing mental disorders: A population-based cohort study' has been provisionally accepted for publication in PLOS Mental Health.

IMPORTANT: The editorial review process is now complete. PLOS will only permit corrections to spelling, formatting or insignificant scientific errors from this point onwards. Requests for major changes, or any which affect the scientific understanding of your work, will cause delays to the publication date of your manuscript.

Best regards,

Hirokazu Taniguchi, Ph.D.

Academic Editor

PLOS Mental Health

Dr. Karli Montague-Cardoso (Executive Editor) Comments:

At PLOS Mental Health, we encourage authors to refrain from using the term 'mental disorder' and would instead encourage the use of 'mental health condition' in the interest of sensitivities across communities and regions. Please consider this when finalising your submission.

Academic Editor's Comments:

Before submitting the completed manuscript, please recalculate the reliability (alpha coefficient) of the Three-Item Loneliness Scale using the sample analyzed in this study (N = 158,970) and include the value in the text, as suggested by Reviewer #1.

Reviewer Comments (if any, and for reference):

Reviewer's Responses to Questions

**Comments to the Author**

1. If the authors have adequately addressed your comments raised in a previous round of review and you feel that this manuscript is now acceptable for publication, you may indicate that here to bypass the “Comments to the Author” section, enter your conflict of interest statement in the “Confidential to Editor” section, and submit your "Accept" recommendation.

Reviewer #1: All comments have been addressed

2. Does this manuscript meet PLOS Mental Health’s publication criteria ? Is the manuscript technically sound, and do the data support the conclusions? The manuscript must describe methodologically and ethically rigorous research with conclusions that are appropriately drawn based on the data presented.

Reviewer #1: Yes

3. Has the statistical analysis been performed appropriately and rigorously?

Reviewer #1: Yes

4. Have the authors made all data underlying the findings in their manuscript fully available (please refer to the Data Availability Statement at the start of the manuscript PDF file)?

Reviewer #1: Yes

5. Is the manuscript presented in an intelligible fashion and written in standard English?

Reviewer #1: Yes

6. Review Comments to the Author

Reviewer #1: Thanks for the detailed reply; it makes sense. One suggestion is that you could calculate Cronbach's alpha yourself based on the data you use, as it varies across datasets.

7. PLOS authors have the option to publish the peer review history of their article (what does this mean? ). If published, this will include your full peer review and any attached files.

**Do you want your identity to be public for this peer review?** For information about this choice, including consent withdrawal, please see our Privacy Policy .

Reviewer #1: No
